# Liposome/gold hybrid nanoparticle encoded with CoQ10 (LGNP-CoQ10) suppressed rheumatoid arthritis via STAT3/Th17 targeting

**Jooyeon Jhun**[1☉], **Jeonghyeon Moon**[2☉], **Jaeyoon Ryu**[1], **Yonghee Shin**[3], **Seangyoun Lee**[1], **Keun-Hyung Cho**[1], **Taewook Kang**[3,4‡]*, **Mi-La Cho**[1,2‡]*, **Sung-Hwan Park**[1‡]*

**1** Rheumatism Research Center, Catholic Research Institute of Medical Science, College of Medicine, The Catholic University of Korea, Seoul, Republic of Korea, **2** Laboratory of Immune Network, Conversant Research Consortium in Immunologic disease, College of Medicine, The Catholic University of Korea, Seoul, Republic of Korea, **3** Department of Chemical and Biomolecular Engineering, Sogang University, Seoul, Republic of Korea, **4** Institute of Integrated Biotechnology, Sogang University, Seoul, Republic of Korea

☉ These authors contributed equally to this work.
‡ TK, MLC and SHP also contributed equally to this work.
* twkang@sogang.ac.kr (TK); iammila@catholic.ac.kr (MLC); rapark@catholic.ac.kr (SHP)

**Data Availability Statement:** All relevant data are within the manuscript.

**Funding:** This research was supported by a grant from the Korea Health Technology R & D Project

## Abstract

Coenzyme Q10 (CoQ10), also known as ubiquinone, is a fat-soluble antioxidant. Although CoQ10 has not been approved as medication by the Food and Drug Administration, it is widely used in dietary supplements. Some studies have shown that CoQ10 has anti-inflammatory effects on various autoimmune disorders. In this study, we investigated the anti-inflammatory effects of liposome/gold hybrid nanoparticles encoded with CoQ10 (LGNP-CoQ10). Both CoQ10 and LGNP-CoQ10 were administered orally to mice with collagen-induced arthritis (CIA) for 10 weeks. The inflammation pathology of joint tissues of CIA mice was then analyzed using hematoxylin and eosin and Safranin O staining, as well as immunohistochemistry analysis. We obtained immunofluorescence staining images of spleen tissues using confocal microscopy. We found that pro-inflammatory cytokines were significantly decreased in LGNP-CoQ10 injected mice. Th17 cell and phosphorylated STAT3-expressed cell populations were also decreased in LGNP-CoQ10 injected mice. When human peripheral blood mononuclear cells (PBMCs) were treated with CoQ10 and LGNP-CoQ10, the IL-17 expression of PBMCs in the LGNP-CoQ10-treated group was significantly reduced. Together, these results suggest that LGNP-CoQ10 has therapeutic potential for the treatment of rheumatoid arthritis.

## 1. Introduction

Rheumatoid arthritis (RA) is an autoimmune disorder that causes long-term and systemic joint inflammation [1]. RA has been reported to lead to bone erosion and cartilage destruction [2]. Although the precise mechanism of RA remains unclear, pro-inflammatory cytokines such as interleukin (IL)-17 are associated with RA pathogenesis [3, 4]. In RA, IL-17 releases CD4+ T cells (helper T 17 [Th17]), which induce an inflammatory response that causes acute

through the Korea Health Industry Development
Institute (KHIDI), funded by the Ministry of Health
& Welfare, Republic of Korea (HI15C1062). The
funders had no role in study design, data collection
and analysis, decision to publish, or preparation of
the manuscript.

**Competing interests:** The authors have declared
that no competing interests exist.

symptoms [5]. Mice with collagen-induced arthritis (CIA) are widely used as an RA disease model [6]. In the present study, we performed clinical and histological analyses using CIA mice to explore RA pathology.

Coenzyme Q10 (CoQ10), which is also known as ubiquinone, is a lipid containing 1,4-benzoquinone and a side chain with 10 isoprenyl subunits in its tail [7]; it is an oil-soluble antioxidant found in most eukaryotic cells [8]. In mitochondria, CoQ10 plays a critical role in producing adenosine triphosphate (ATP) via the oxidative phosphorylation pathway [9, 10]. Recently, CoQ10 has been reported to have anti-inflammatory functions and exhibit therapeutic effects for immune disorders [11, 12]. Some studies have demonstrated therapeutic effects of CoQ10 against RA through the regulation of immune process factors such as tumor necrosis factor (TNF)-α and IL-6 [13, 14]. Our previous studies showed that CoQ10 inhibited autoimmune disease pathogenesis [15, 16]. We also demonstrated that CoQ10 suppresses Th17 cells in experimental autoimmune arthritis mice via inhibition of the signal transducer and activator of transcription (STAT3) signaling pathway [17, 18].

Thus, CoQ10 confers various benefits against immune diseases; however, it is a water-insoluble component, such that its absorption in the small intestine is inefficient [19]. Therefore, we used liposome/gold hybrid nanoparticles encoded with CoQ10 (LGNP-CoQ10) to improve CoQ10 absorption [20]. This hybrid system is dependent on temperature, pH, and light activation and therefore has site-specific drug delivery functions and is safe in cells [21]. Some studies have demonstrated its therapeutic effects on immune cells such as dendritic cells and macrophages [22, 23].

In the present study, we hypothesized that LGNP-CoQ10 suppresses the pathology of experimental autoimmune arthritis. We established liposomes to treat CIA mice with different levels of CoQ10 (no CoQ10, CoQ10 alone, and CoQ10/gold hybrid nanoparticles) for 10 weeks. We then examined whether these liposomes had therapeutic effects in CIA mice.

## 2. Materials and methods

### 2.1 Animals

We maintained 7-week-old male DBA/1J mice (Orient Bio, Gyeonggi-do, Korea) under specific pathogen-free conditions. The mice were fed standard laboratory mouse chow (Ralston Purina, St. Louis, MO, USA) and water *ad libitum*; they were housed (five mice per cage) in a room under controlled temperature (21–22˚C) and lighting (12-h light/12-h dark cycle) conditions. The Animal Care Committee of The Catholic University of Korea approved the experimental protocol. All experimental procedures were evaluated and conducted in accordance with the protocols approved by the Animal Research Ethics Committee at the Catholic University of Korea (ID number: 2017-0104-01). All procedures performed in this study followed the ethical guidelines for animal use.

### 2.2 Formation of LGNP-CoQ10

A chloroform solution containing 500 μg saturated neutral phospholipids (1,2-distearoyl-sn-glycero-3-phosphocholine [DSPC]) was dropped into a glass vial. Next, a lipid film was formed by evaporating the chloroform solution under an $N_2$ stream. The film was maintained in a vacuum for longer than 1 h and then hydrated at 55˚C by adding 0.5 mL ascorbic acid solution ($C_6H_8O_6$, 600 mM, pH 5) and 0.5 mL CoQ10 solution. To obtain unilamellar liposomes, as-prepared multilamellar liposomes were extruded through a polycarbonate membrane (pore size: 100 nm) 20 times at 60˚C using a mini-extruder. Unilamellar liposomes were purified by centrifugation and washed three times with deionized water to remove residues. Finally, the liposomes encoded with ascorbic acid and CoQ10 were re-suspended using a gold precursor

aqueous solution (HAuCl$_4$·3H$_2$O, tetrachloroauric acid trihydrate, 200 μM) and kept over-night in an agitating incubator at room temperature [24]. The LGNP-CoQ10 structure is shown in Fig 1A.

## 2.3 Arthritis induction and treatment

CIA was induced in DBA1/J mice (n = 5). The experiment was performed in triplicate. Type II collagen (CII) was dissolved overnight in 0.1 N acetic acid (4 mg/mL) with gentle rotation at 4˚C. Male DBA/1J mice were immunized intra-dermally at the base of the tail with 100 μg chicken CII (Chondrex, Inc., Remosa, WA, USA) in complete Freund's adjuvant (Chondrex Inc.). In experiments conducted to investigate preventive effects, mice were boosted with 100 μg CII emulsified with incomplete Freund's adjuvant (Chondrex Inc.), injected intrader-mally at the base of the tail on day 17 after primary immunization. These arthritis model mice

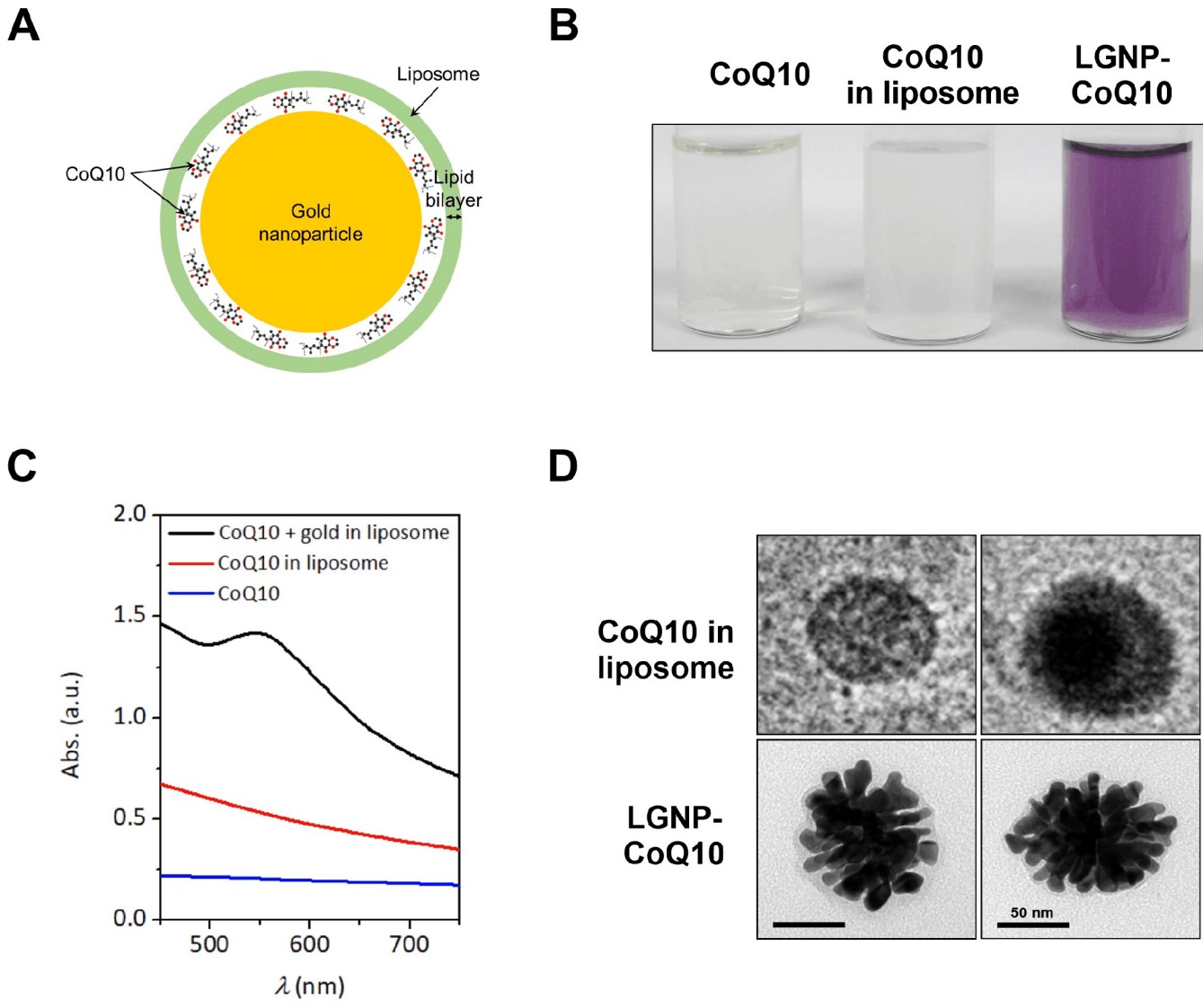

**Fig 1. Formation of liposome/gold nanoparticles encoded with CoQ10 (LGNP-CoQ10).** (A) Structure of the LGNP-CoQ10. (B and C) The cloudy color and ultraviolet-visible (UV-vis) absorbance of CoQ10, CoQ10 liposome, and CoQ10+gold liposome solutions are presented. (D) The morphology of liposomes that contained only CoQ10 and gold nanoparticles + CoQ10 was assessed by transmission electron microscopy (TEM). Scale bars = 50 nm.

were treated orally with Gold Lipo CoQ10 (0.1 mg/mouse) or CoQ10 (0.1 mg/mouse) starting on day 17 after the first immunization. The mice were examined visually twice weekly for the appearance of arthritis in the peripheral joints. All mice were sacrificed on week 10 for histological analyses of splenocytes and determination of protein expression.

## 2.4 Clinical assessment of arthritis

Arthritis severity was recorded using the mean arthritis index on a scale of 0 to 4, as follows: (0) no evidence of erythema or swelling; (1) erythema and mild swelling confined to the midfoot (tarsals) or ankle joint; (2) erythema and mild swelling extending from the ankle to the midfoot; (3) erythema and moderate swelling extending from the ankle to the metatarsal joints; and (4) erythema and severe swelling encompassing the ankle, foot, and digits. Arthritis severity was determined as the sum of the scores of all legs, as assessed by two independent observers with no knowledge of the experimental groups.

## 2.5 Histological analyses

Mouse joint tissues were fixed in 4% paraformaldehyde (Sigma-Aldrich, St. Louis, MO, USA), decalcified in histological decalcifying agent (Calci-Clear Rapid; National Diagnostics, Atlanta, GA, USA), trimmed, and embedded in paraffin wax. Sections (7 μm) were prepared and stained with hematoxylin (YD Diagnostics, Yongin, Korea), eosin (Muto Pure Chemicals Co., Ltd., Tokyo, Japan), and Safranin O (Sigma-Aldrich). Cartilage damage was scored as described previously [25].

## 2.6 Immunohistopathological analyses

Joint tissues were first incubated with primary antibodies against IL-1β (R & D Systems, Minneapolis, MN, USA), IL-6 (R & D Systems), TNF-α (R & D Systems), IL-17 (R & D Systems), and RANK ligand (RANKL; R & D Systems) overnight at 4°C. Samples were incubated with a biotinylated secondary antibody, followed by incubation with a streptavidin–peroxidase complex for 1 h. Samples were then developed using chromogen 3,3′-diaminobenzidine (Thermo Scientific, Rockford, IL, USA). The sections were examined using a photomicroscope (Olympus, Tokyo, Japan).

## 2.7 Confocal microscopy of immunostaining

Spleen tissues were obtained at 10 weeks after the first immunization. Tissues were snap-frozen in liquid nitrogen and stored at –80°C. Tissue sections were fixed in 4% paraformaldehyde and stained with phycoerythrin (PE)-conjugated rat monoclonal anti-CD4, fluorescein isothiocyanate (FITC)-conjugated rat monoclonal anti-IL-17, FITC-conjugated mouse monoclonal anti-pSTAT3 705, and FITC-conjugated mouse monoclonal anti-pSTAT3 727 (all from eBiosciences, San Diego, CA, USA). Stained sections were analyzed microscopically (LSM510Meta; Carl Zeiss, Oberkochen, Germany).

## 2.8 Cell culture

Splenocytes were prepared from the spleens of normal C57BL/6 mice. Splenocytes were maintained in RPMI1640 medium supplemented with 5% fetal bovine serum (Gibco, Grand Island, NY, USA) and stimulated with anti-CD3 (0.5 μg/mL; BD Biosciences, San Jose, CA, USA) for 3 days and subjected to enzyme-linked immunosorbent assay (ELISA).

## 2.9 Statistical analyses

Statistical analyses were conducted using the nonparametric Mann–Whitney U test for comparisons between two groups, and one-way analysis of variance (ANOVA) with Bonferroni's post hoc test for multiple comparisons. We used the GraphPad Prism v. 5.01 software (GraphPad Software Inc., San Diego, CA, USA) for all analyses. The threshold for statistical significance was $P < 0.05$. All data are presented as means ± standard deviation (SD).

# 3. Results

## 3.1 Formation of LGNP-CoQ10

Liposomes encoded with a reducing agent (i.e., ascorbic acid) and coenzyme Q10 (CoQ10) were prepared through the extrusion of multilamellar liposomes (see Methods). To form gold nanoparticles within the liposomes, gold precursor solution was added into the as-prepared liposomes (Fig 1A). The cloudy color of the solution gradually changed to violet over time (Fig 1B). The solution was sampled for ultraviolet-visible absorbance and transmission electron microscopy (TEM) measurements. The absorbance spectrum clearly exhibited a characteristic surface plasmon resonance band at 545 nm (Fig 1C). To characterize the liposome, TEM measurements were performed; TEM images showed round-shaped structures without any ruptures (Fig 1D). Representative TEM images showed nanoparticles surrounded by a lipid layer, indicating the formation of gold nanoparticles within liposomes encoded with CoQ10.

## 3.2 The pathogenesis of rheumatoid arthritis was suppressed by LGNP-CoQ10 in CIA mice

To determine whether LGNP-CoQ10 had therapeutic effects in CIA mice, liposomes were injected into the mice for 10 weeks. Pathology scores and incidence of arthritis were significantly reduced in CoQ10- and LGNP-CoQ10-injected mice (Fig 2A). Although it was effective in both, the treatment of LGNP-CoQ10 had a higher therapeutic effect on RA. Mouse joint tissues were monitored for tissue disruption following hematoxylin and eosin and Safranin O staining (Fig 2B). Histological analysis showed that the average scores of bone damage, cartilage damage, and inflammation were notably reduced in the LGNP-CoQ10-injected group.

## 3.3 Expression of pro-inflammatory cytokines was decreased in joints of LGNP-CoQ10-treated mice

We obtained mouse joint tissues at the end of the experiments. Pro-inflammatory cytokines such as IL-1β, IL-6, IL-17 and TNF-α, and an osteoclast differentiation marker, RANKL, were observed in the joint tissues by immunohistochemistry (Fig 3). The tissue immunohistochemistry images of CoQ10-injected mice showed a significant reduction of the pro-inflammatory cytokines and RANKL. Besides, the joint tissues which were treated the LGNP-CoQ10 were more decreased the pro-inflammatory cytokines and RANKL than only the CoQ10-treated group. This result suggested that the LGNP-CoQ10 system is more effective using CoQ10 alone.

## 3.4 Expression of IL-17 and phosphorylated STAT3 was reduced by LGNP-CoQ10 in spleen CD4+ T cells

After the mice were sacrificed, we obtained spleen tissues and performed immunofluorescence staining (Fig 4). In spleen tissues, Th17 cells were decreased following CoQ10 treatment, and more significantly following LGNP-CoQ10 treatment. Besides, the spleen tissue

**A**

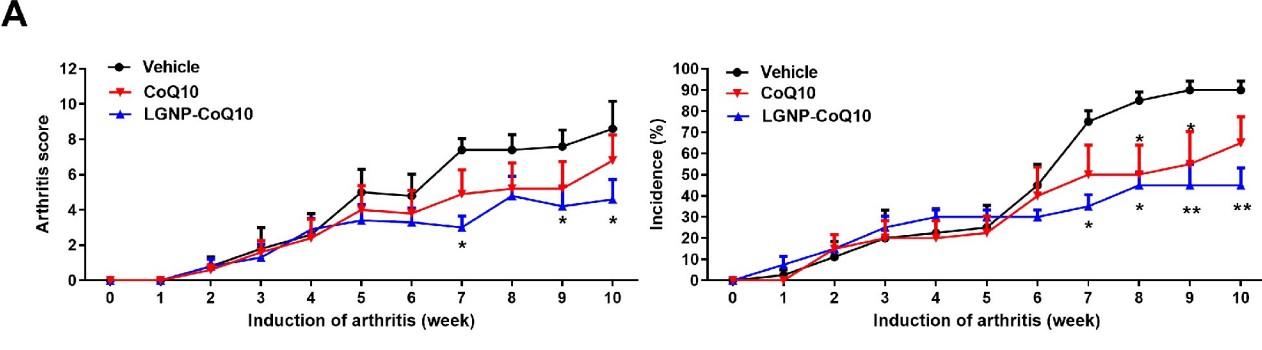

**B**

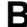
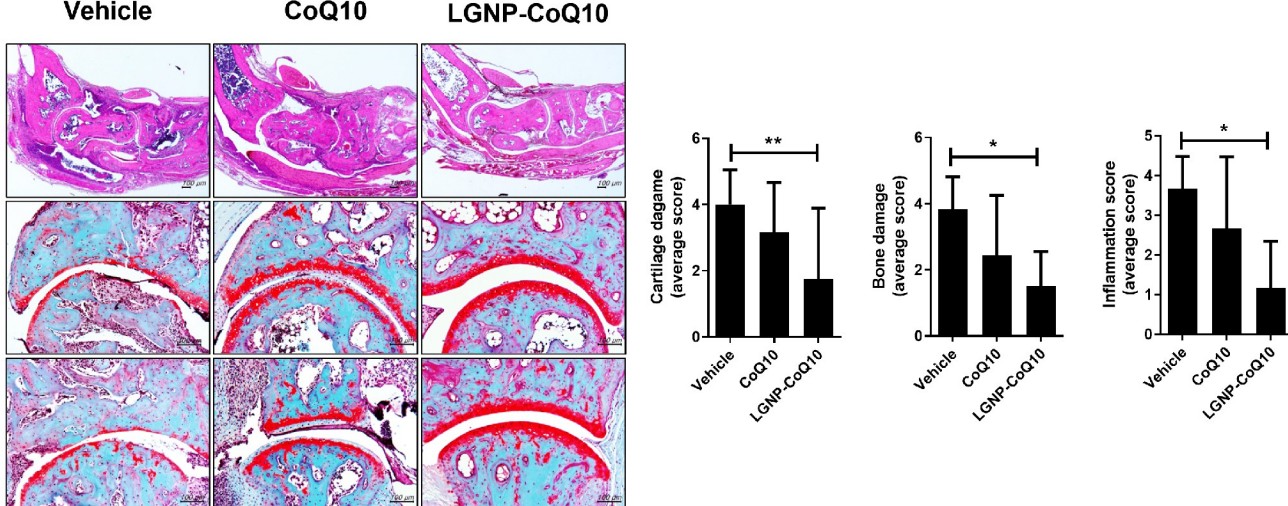

**Fig 2. Pathology scores of CoQ10 and LGNP-CoQ10-injected mice.** (A) The arthritis score and incidence of CIA mice in each group are shown. (B) Joint tissues of CIA mice were stained with hematoxylin and eosin as well as Safranin O. *P < 0.05; **P < 0.01. Scale bars = 100 μm.

immunofluorescence images showed a reduction in phosphorylated STAT3 among CoQ10-treated mice. This data suggested that the CoQ10 and LGNP-CoQ10 suppressed the IL-17 producing cells in the spleen tissue.

### 3.5 IL-17 levels decreased in peripheral blood mononuclear cells following treatment with LGNP-CoQ10

To investigate the effects of CoQ10 and LGNP-CoQ10 treatment in reducing IL-17 expression, we treated human peripheral blood mononuclear cells (PBMCs) with CoQ10 and LGNP-CoQ10 under anti-CD3 conditions. IL-17 levels in the culture supernatants were detected by ELISA, and found to be significantly decreased in LGNP-CoQ10-treated PBMCs (Fig 5).

## 4. Discussion

Although CoQ10 has been studied as a powerful antioxidant, there is accumulating evidence that it has anti-inflammatory functions. In the present study, we demonstrated that CoQ10 has

## A

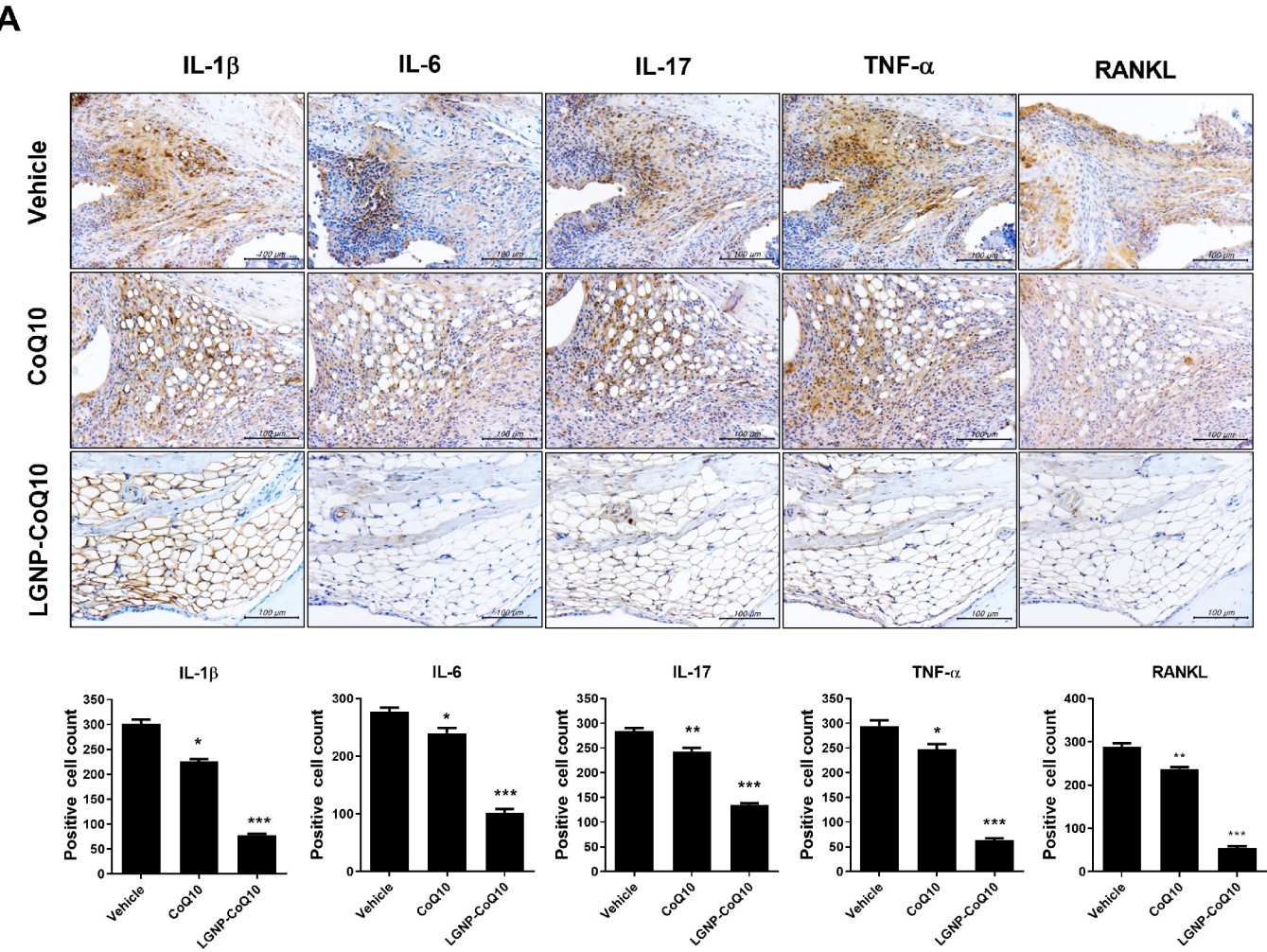

**Fig 3. Immunohistochemistry of joint tissues in vehicle-, CoQ10-, and LGNP-CoQ10-injected mice.** $^*P < 0.05$; $^{**}P < 0.01$; and $^{***}P < 0.001$. Scale bars = 100 μm.

anti-inflammatory effects on experimental autoimmune arthritis. We also investigated the therapeutic functions of LGNP-CoQ10.

In our previous studies, we demonstrated that CoQ10 has anti-inflammatory functions through the suppression of Th17 cells [15–17]. In particular, we showed that CoQ10 inhibited Th17 cells through the suppression of phosphorylated STAT3 (p-STAT3) production [18]; p-STAT3 is the activated form of STAT3, and is known to increase levels of pro-inflammatory cytokines such as TNF-α, IL-6, and IL-1β [26].

CoQ10 confers various benefits and is found in some foods such as sardines, mackerels, and green leafy vegetables including soybeans, peanuts, and beef liver. Therefore, humans typically ingest 3–5 mg of CoQ10 per day in food [27]. Despite the intake of high doses of CoQ10, its absorption efficiency in the gastrointestinal tract is poor [28]. CoQ10 has a biopharmaceutics classification system (BCS) index of 2 or 4, which are typical values for low-solubility drugs [29]. Therefore, we applied a gold/liposome nanoparticle hybrid system to improve CoQ10 absorption in this study.

We established liposome structures including CoQ10, as well as CoQ10/gold nanoparticle hybrids. Our previous studies demonstrated that the LGNP system has higher absorption

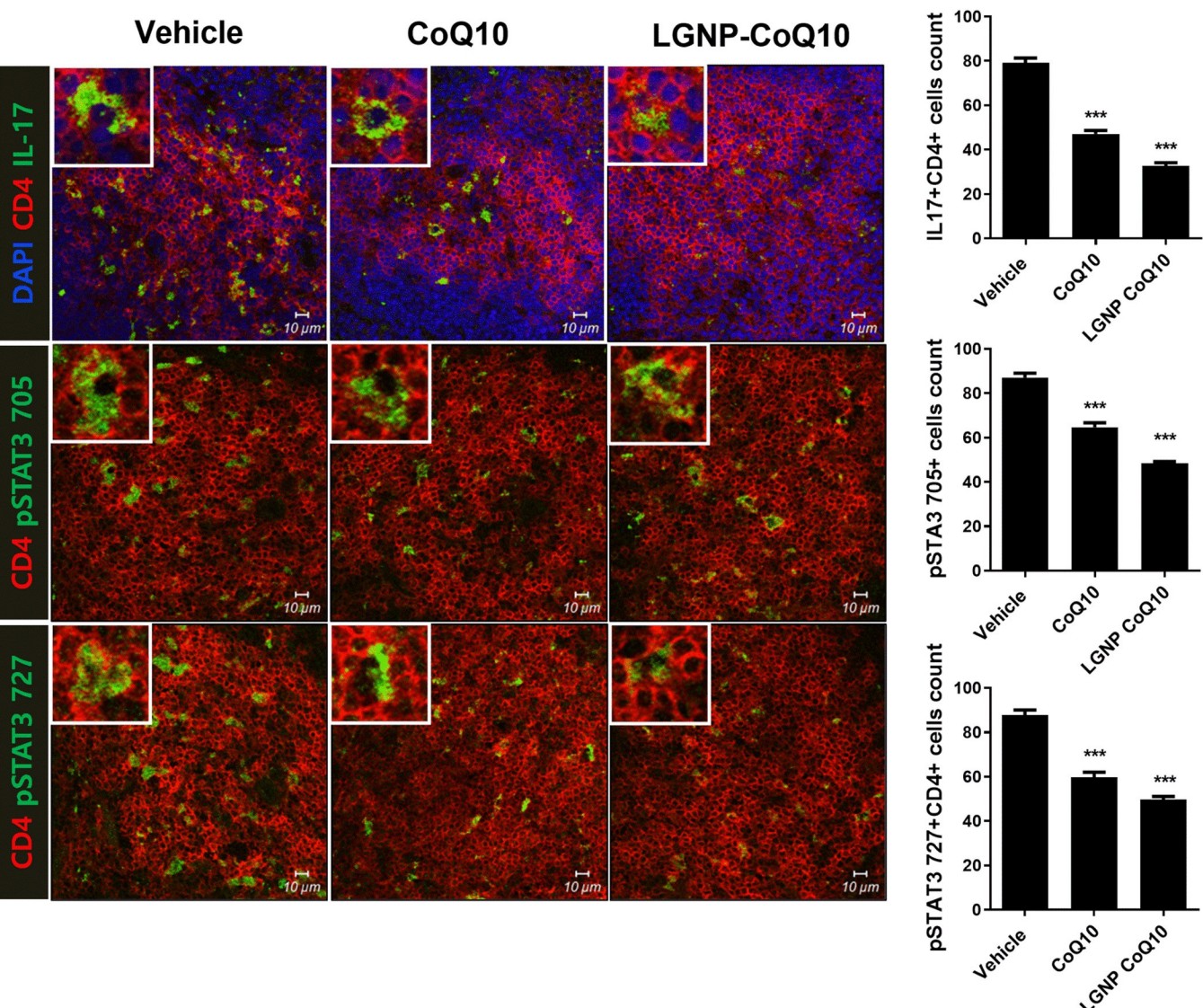

**Fig 4. Immunofluorescence of spleen tissues.** (A and B) Th17 cells and phosphorylated STAT3 expression in the spleens of treatment with vehicle, CoQ10 and LGNP-CoQ10 were detected by confocal microscopy. *P < 0.05; **P < 0.01; and ***P < 0.001. Scale bars = 10 μm.

efficiency [30, 31]. LGNP system facilitates endocytosis by wrapping the particles in a phospho-lipid bilayer which is the same with the cell's surface, helping the cells absorb the particles. These liposomes were injected orally in CIA mice. We found that CoQ10 or CoQ10/gold nano-particles suppressed the pathogenesis of experimental RA in the animal model. RA pathology was significantly decreased in LGNP-CoQ10-treated mice; however, there were no significant differences in the group treated with CoQ10 alone. Moreover, inflammation, bone damage, and cartilage damage scores were notably reduced in the LGNP-CoQ10-injected mice. Pro-inflam-matory cytokines (IL-1β, IL-6, IL-17, TNF-α, and RANKL) were decreased in CIA mice treated with LGNP-CoQ10. Interestingly, Th17 cells decreased in the spleen tissue of CoQ10/gold nanoparticle-injected mice. We also investigated p-STAT3 levels in the spleen, and found that p-STAT3 was reduced in CD4+ T cells. A previous study showed that the increase of ROS pro-motes Th17 differentiation and IL-17 production through the activation of RORγt and STAT3

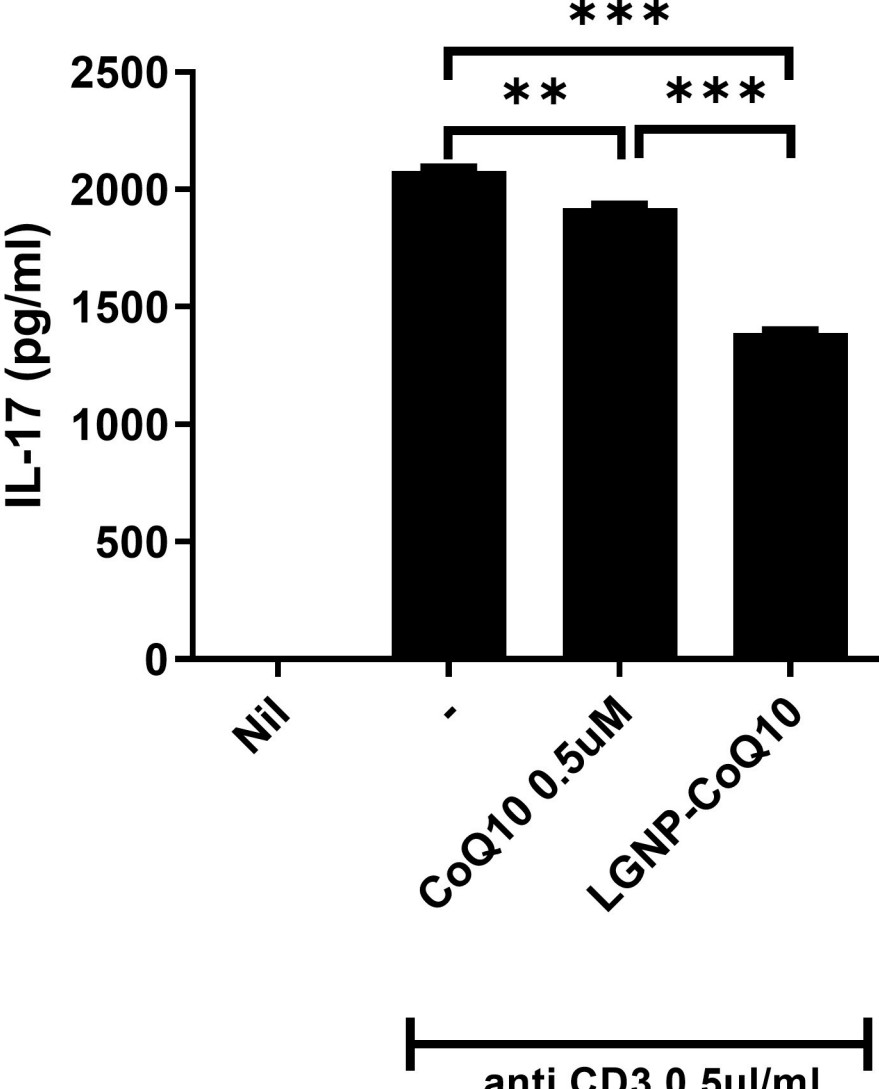

**Fig 5. IL-17 levels were detected by ELISA in human peripheral blood mononuclear cells (PBMC) in anti-CD3 conditions.** The isolated PBMC was seeded and treated the 0.5μM of CoQ10 and LGNP-CoQ10 for 3 days respectively. **P < 0.01 and ***P < 0.001.

pathway [32]. CoQ10 is a reactive oxygen species (ROS) scavenger. Therefore, CoQ10 can directly or indirectly regulate Th17 and IL-17 as ROS scavengers through the STAT3 pathway. Together, these data suggest that the combination of CoQ10 and gold nanoparticles has more effective anti-inflammatory effects than liposome structures containing only CoQ10.

The results of the present study demonstrate that CoQ10 treatment suppressed the pathogenesis of RA in CIA mice. In particular, LGNP-CoQ10 reduced IL-17 levels through the inhibition of p-STAT3. Together, these results show that LGNP-CoQ10 has therapeutic potential for treating RA.

## 5 Conclusion

The hybrid nanoparticles encoded with Coenzyme Q10 (LGNP-CoQ10) has more effective functions than CoQ10 in rheumatoid arthritis. LGNP-CoQ10 has therapeutic potential for the

treatment of rheumatoid arthritis. It was reduced IL-17 level through the inhibition of phosphorylated STAT3.

## Author Contributions

**Conceptualization:** Jooyeon Jhun, Jeonghyeon Moon, Taewook Kang, Mi-La Cho, Sung-Hwan Park.

**Data curation:** Jooyeon Jhun, Jeonghyeon Moon.

**Formal analysis:** Jooyeon Jhun, Jeonghyeon Moon, Yonghee Shin, Seangyoun Lee.

**Funding acquisition:** Jaeyoon Ryu.

**Methodology:** Yonghee Shin, Keun-Hyung Cho, Taewook Kang.

**Supervision:** Taewook Kang, Mi-La Cho, Sung-Hwan Park.

**Writing – original draft:** Jooyeon Jhun, Jeonghyeon Moon.

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
