## [Decision Letter · Decision Letter 0]

8 Sep 2020

PONE-D-20-25707

Liposome/gold hybrid nanoparticle encoded with CoQ10 (LGNP-CoQ10) suppressed rheumatoid arthritis via STAT3/Th17 targeting

PLOS ONE

Dear Dr. Cho,

Thank you for submitting your manuscript to PLOS ONE. After careful consideration, we feel that it has merit but does not fully meet PLOS ONE’s publication criteria as it currently stands. Therefore, we invite you to submit a revised version of the manuscript that addresses the points raised during the review process.

Please address all critiques raised by the reviewers. Particularly, reviewer #1 raised a few critical comments regarding the role of the nanoparticle on Th17 cell responses. Other minor comments should also be addressed in the revised manuscript.

We look forward to receiving your revised manuscript.

Kind regards,

Yeonseok Chung

Academic Editor

PLOS ONE

Journal Requirements:

2. Your ethics statement must appear in the Methods section of your manuscript. If your ethics statement is written in any section besides the Methods, please move it to the Methods section and delete it from any other section. Please also ensure that your ethics statement is included in your manuscript, as the ethics section of your online submission will not be published alongside your manuscript.

Reviewers' comments:

Reviewer's Responses to Questions

**Comments to the Author**

1. Is the manuscript technically sound, and do the data support the conclusions?

Reviewer #1: Partly

Reviewer #2: Yes

2. Has the statistical analysis been performed appropriately and rigorously? 

Reviewer #1: Yes

Reviewer #2: Yes

3. Have the authors made all data underlying the findings in their manuscript fully available?

Reviewer #1: Yes

Reviewer #2: Yes

4. Is the manuscript presented in an intelligible fashion and written in standard English?

Reviewer #1: No

Reviewer #2: Yes

5. Review Comments to the Author

Reviewer #1: In this manuscript, authors demonstrated the enhanced anti-inflammatory effect of LGNP-CoQ10 compared to CoQ10 in collagen induced arthritis model (CIA). Oral administration of LGNP-CoQ10 ameliorated the pathogenesis of CIA by decreasing inflammatory cytokines including IL-1β, IL-6, IL-17, and TNF-α. The therapeutic effect of LGNP-CoQ10 is impressive. However, several concerns should be addressed to convincingly describe the author’s novel findings.

Major points

- The data showed that LGNP-CoQ10 inhibits CIA through IL-17 related pathways. However, it is not clear that LGNP-CoQ10 reduces IL-17 production from CD4 T cells. As authors titled this article that ‘Liposome/gold hybrid nanoparticle encoded with CoQ10 (LGNP-CoQ10) suppressed rheumatoid arthritis via STAT3/‘Th17’ targeting’, please show the LGNP-CoQ10 mediated IL-17 inhibition in CD4 T cells and other IL-17–producing cells.

- How does LGNP-CoQ10 reduces IL-17 levels? For example, LGNP-CoQ10 inhibits induction of IL-17–producing cells or decreases the maintenance of established IL-17–producing cells?

Minor points

- Please describe the meaning of each data more in detail including Figure 3 in the result section.

- Figure 4B is not marked in the Figure 4.

Reviewer #2: This study evaluated the anti-inflammatory effect of CoQ in liposome/gold hybrid nanoparticle (LGNP-CoQ10) in the animal model of rheumatoid arthritis. Although the therapeutic potential of LGNP-CoQ10 against rheumatoid arthritis seems promising according to the results in the study, the current manuscript has too many typos and errors and did not extensively describe or interpret the data. Please find my comments as below.

Materials and Methods

1. In the Confocal microscopy of immunostaining section, please provide the clone name for the antibodies used in the study.

2. In the Cell culture section, C57NL/6? Is this a typo of C57BL/6?

Results

3. In the “Formation of LGNP-CoQ10”, no description about Figure 1A

4. In the “The pathogenesis of rheumatoid arthritis was suppressed by LGNP-CoQ10 in CIA mice”, the authors should compare the results of all tested experimental groups in the figure. They did not describe the result of CoQ10 in Figure 2A. Also, in the last sentence of this paragraph, the authors did not interpret the data of Figure 3.

5. In the “Expression of IL-17 and phosphorylated STAT3…”, there is no B in Figure 4B.

Figures

6. Please add B in Figure 4

7. Please remove A in the figure 5.

Figure legends

8. In the legend of Figure 4, please add CoQ10 in the second sentence.

9. In the legend of Figure 5, please describe the experimental condition in more detail including the addition of CoQ10 and NGNP-CoQ10.

Discussion

10. In the fourth paragraph, the authors demonstrated that “We found that only liposomes containing CoQ10 or CoQ10/gold nanoparticles suppressed the pathogenesis of experimental RA in the animal model.”. Throughout the study except Figure 1, the authors compared LGNP-CoQ10 with CoQ10 alone (or vehicle) but they did not include CoQ10 in liposome as an experimental group. Please make it clear whether CoQ group in the Figure 2-5 means CoQ10 itself or CoQ10 in liposome.

11. Please discuss how LGNP can improve the solubility or absorption of CoQ10 in vivo.

12. Is the effect of CoQ10 on suppression of Stat3 phosphorylation direct or indirect? Please explain the mechanism by which CoQ10 modulate the Stat3 phosphorylation in Th17 cells.

6. PLOS authors have the option to publish the peer review history of their article (what does this mean?). If published, this will include your full peer review and any attached files.

Reviewer #1: No

Reviewer #2: No

---

## [Author Response · Author response to Decision Letter 0]

6 Oct 2020

Reviewer #1: In this manuscript, authors demonstrated the enhanced anti-inflammatory effect of LGNP-CoQ10 compared to CoQ10 in collagen induced arthritis model (CIA). Oral administration of LGNP-CoQ10 ameliorated the pathogenesis of CIA by decreasing inflammatory cytokines including IL-1β, IL-6, IL-17, and TNF-α. The therapeutic effect of LGNP-CoQ10 is impressive. However, several concerns should be addressed to convincingly describe the author’s novel findings.

Major points

- The data showed that LGNP-CoQ10 inhibits CIA through IL-17 related pathways. However, it is not clear that LGNP-CoQ10 reduces IL-17 production from CD4 T cells. As authors titled this article that ‘Liposome/gold hybrid nanoparticle encoded with CoQ10 (LGNP-CoQ10) suppressed rheumatoid arthritis via STAT3/‘Th17’ targeting’, please show the LGNP-CoQ10 mediated IL-17 inhibition in CD4 T cells and other IL-17–producing cells.

Thank you for your comments. As the reviewer pointed, the authors investigated the population of Th17 using flow cytometry. Although the population of Th17 cells of CoQ10 and LGNP-CoQ10 injected group tended to decrease, however, there were no significant differences. So, we didn’t put the data in the manuscript. Instead of the FACS data, we added the immunofluorescence images of spleen tissues (Figure 4) and serum ELISA data (Figure 5) in this study. 

- How does LGNP-CoQ10 reduces IL-17 levels? For example, LGNP-CoQ10 inhibits induction of IL-17–producing cells or decreases the maintenance of established IL-17–producing cells?

As the reviewer pointed, the accurate mechanism of reduction of IL-17 by LGNP-CoQ10 was not conducted in this paper. However, our previous studies confirmed the Coenzyme Q10 has a therapeutic effect on rheumatoid arthritis through the regulation of IL-17 level (Lee SY et al. A Combination with Probiotic Complex, Zinc, and Coenzyme Q10 Attenuates Autoimmune Arthritis by Regulation of Th17/Treg Balance. J Med Food. 2018; Lee SH et al. Coenzyme Q10 Exerts Anti-Inflammatory Activity and Induces Treg in Graft Versus Host Disease. J Med Food. 2016; Jhun J et al. Combination therapy with metformin and coenzyme Q10 in murine experimental autoimmune arthritis. Immunopharmacol Immunotoxicol. 2016). besides, coenzyme Q10 suppressed the differentiation of Th17 cells and osteoclast, it also inhibited the STAT3 signaling pathway in the autoimmune disease mouse model. (Jhun J et al. Coenzyme Q10 suppresses Th17 cells and osteoclast differentiation and ameliorates experimental autoimmune arthritis mice. Immunol Lett. 2015; Lee SY et al. Coenzyme Q10 Inhibits Th17 and STAT3 Signaling Pathways to Ameliorate Colitis in Mice. J Med Food. 2017.) LGNP-CoQ10 system was used as an improvement process to increase the absorption rate of CoQ10. For this reason, the mechanism for reducing IL-17 by LGNP-CoQ10 is expected to be similar to previous studies. 

Minor points

- Please describe the meaning of each data more in detail including Figure 3 in the result section.

As the reviewer commented, the author added more detail explanation in the manuscript. “We obtained mouse joint tissues at the end of the experiments. Pro-inflammatory cytokines such as IL-1β, IL-6, IL-17 and TNF-α, and an osteoclast differentiation marker, RANKL, were observed in the joint tissues by immunohistochemistry (Figure 3). The tissue immunohistochemistry images of CoQ10-injected mice showed a significant reduction of the pro-inflammatory cytokines and RANKL. Besides, the joint tissues which were treated the LGNP-CoQ10 were more decreased the pro-inflammatory cytokines and RANKL than only the CoQ10-treated group. This result suggested that the LGNP-CoQ10 system is more effective using CoQ10 alone.”

- Figure 4B is not marked in the Figure 4.

As the reviewer pointed, the author edited the manuscript and the figure 4. “After the mice were sacrificed, we obtained spleen tissues and performed immunofluorescence staining (Figure 4). In spleen tissues, Th17 cells were decreased following CoQ10 treatment, and more significantly following LGNP-CoQ10 treatment. Besides, the spleen tissue immunofluorescence images showed a reduction in phosphorylated STAT3 among CoQ10-treated mice. This data suggested that the CoQ10 and LGNP-CoQ10 suppressed the IL-17 producing cells in the spleen tissue.”

 

Reviewer #2: This study evaluated the anti-inflammatory effect of CoQ in liposome/gold hybrid nanoparticle (LGNP-CoQ10) in the animal model of rheumatoid arthritis. Although the therapeutic potential of LGNP-CoQ10 against rheumatoid arthritis seems promising according to the results in the study, the current manuscript has too many typos and errors and did not extensively describe or interpret the data. Please find my comments as below.

Materials and Methods

1. In the Confocal microscopy of immunostaining section, please provide the clone name for the antibodies used in the study.

Thank you for your comments. As the reviewer pointed, the authors added the clone name of antibodies in the manuscript. “Tissue sections were fixed in 4% paraformaldehyde and stained with phycoerythrin (PE)-conjugated rat monoclonal anti-CD4, fluorescein isothiocyanate (FITC)-conjugated rat monoclonal anti-IL-17, FITC-conjugated mouse monoclonal anti-pSTAT3 705, and FITC-conjugated mouse monoclonal anti-pSTAT3 727 (all from eBiosciences, San Diego, CA, USA).”

2. In the Cell culture section, C57NL/6? Is this a typo of C57BL/6?

As the reviewer pointed, the authors edited the typo in the manuscript. “Splenocytes were prepared from the spleens of normal C57BL/6 mice.”

Results

3. In the “Formation of LGNP-CoQ10”, no description about Figure 1A

As the reviewer pointed, the authors edited the manuscript. “To form gold nanoparticles within the liposomes, gold precursor solution was added into the as-prepared liposomes (Figure 1A).”

4. In the “The pathogenesis of rheumatoid arthritis was suppressed by LGNP-CoQ10 in CIA mice”, the authors should compare the results of all tested experimental groups in the figure. They did not describe the result of CoQ10 in Figure 2A. Also, in the last sentence of this paragraph, the authors did not interpret the data of Figure 3.

As the reviewer pointed, the authors edited the manuscript and added more detail explanation. “Pathology scores and incidence of arthritis were significantly reduced in CoQ10- and LGNP-CoQ10-injected mice (Figure 2A). Although it was effective in both, the treatment of LGNP-CoQ10 had a higher therapeutic effect on RA.” and “The tissue immunohistochemistry images of CoQ10-injected mice showed a significant reduction of the pro-inflammatory cytokines and RANKL. Besides, the joint tissues which were treated the LGNP-CoQ10 were more decreased the pro-inflammatory cytokines and RANKL than only the CoQ10-treated group. This result suggested that the LGNP-CoQ10 system is more effective using CoQ10 alone.”

5. In the “Expression of IL-17 and phosphorylated STAT3…”, there is no B in Figure 4B.

As the reviewer pointed, the authors edited the manuscript and figure. “After the mice were sacrificed, we obtained spleen tissues and performed immunofluorescence staining (Figure 4). In spleen tissues, Th17 cells were decreased following CoQ10 treatment, and more significantly following LGNP-CoQ10 treatment. Besides, the spleen tissue immunofluorescence images showed a reduction in phosphorylated STAT3 among CoQ10-treated mice. This data suggested that the CoQ10 and LGNP-CoQ10 suppressed the IL-17 producing cells in the spleen tissue.”

Figures

6. Please add B in Figure 4

As the reviewer pointed, the authors edited the Figure 4.

7. Please remove A in the figure 5.

As the reviewer pointed, the authors edited the Figure 5.

Figure legends

8. In the legend of Figure 4, please add CoQ10 in the second sentence.

As the reviewer commented, the authors added “CoQ10” in the legend of Figure 4.

9. In the legend of Figure 5, please describe the experimental condition in more detail including the addition of CoQ10 and NGNP-CoQ10.

As the reviewer pointed, the authors added more detail explanation. “IL-17 levels were detected by ELISA in human peripheral blood mononuclear cells (PBMC) in anti-CD3 conditions. The isolated PBMC was seeded and treated the 0.5µM of CoQ10 and LGNP-CoQ10 for 3 days respectively.”

Discussion

10. In the fourth paragraph, the authors demonstrated that “We found that only liposomes containing CoQ10 or CoQ10/gold nanoparticles suppressed the pathogenesis of experimental RA in the animal model.”. Throughout the study except Figure 1, the authors compared LGNP-CoQ10 with CoQ10 alone (or vehicle) but they did not include CoQ10 in liposome as an experimental group. Please make it clear whether CoQ group in the Figure 2-5 means CoQ10 itself or CoQ10 in liposome.

We mean it is the CoQ10 exclusive use group. So, the authors edited the manuscript as the reviewer pointed. “We found that CoQ10 or CoQ10/gold nanoparticles suppressed the pathogenesis of experimental RA in the animal model.”

11. Please discuss how LGNP can improve the solubility or absorption of CoQ10 in vivo.

Our previous studies demonstrated that the liposomal system with gold nanoparticles has higher absorption efficiency (Jin-Ho Lee et al. General and programmable synthesis of hybrid liposome/metal nanoparticles. Sci Adv. 2016; Youngjae Lee et al. Phase transfer-driven rapid and complete ligand exchange for molecular assembly of phospholipid bilayers on aqueous gold nanocrystals. Chem Commun. 2019.). This system facilitates endocytosis by wrapping the particles in a phospholipid bilayer like the cell’s surface, helping the cells to absorb the particles. The authors added the information in the discussion part of the manuscript. “Our previous studies demonstrated that the LGNP system has higher absorption efficiency [30, 31]. LGNP system facilitates endocytosis by wrapping the particles in a phospholipid bilayer which is the same with cell’s surface, helping the cells absorb the particles.”

12. Is the effect of CoQ10 on suppression of Stat3 phosphorylation direct or indirect? Please explain the mechanism by which CoQ10 modulate the Stat3 phosphorylation in Th17 cells.

As the reviewer pointed, the detailed mechanism of the regulation effect of CoQ10 in Th17 cells is not described in this manuscript. Although our previous studies showed the therapeutic effect of CoQ10 in autoimmune diseases, the accurate mechanism of CoQ10 or LGNP system was not investigated (Jhun J et al. Coenzyme Q10 suppresses Th17 cells and osteoclast differentiation and ameliorates experimental autoimmune arthritis mice. Immunol Lett. 2015; Lee SY et al. Coenzyme Q10 Inhibits Th17 and STAT3 Signaling Pathways to Ameliorate Colitis in Mice. J Med Food. 2017.). For this reason, we are trying to discover the mechanism. We intend to present it in a further study. However, we can infer the process with the following study (Eric V. Dang. Control of TH17/Treg Balance by Hypoxia-Inducible Factor 1. Cell. 2011.). In this study, the increase of ROS promotes Th17 differentiation and IL-17 production through the activation of RORγt and STAT3. Coenzyme Q10 is a ROS scavenger. So, CoQ10 has a regulatory effect on Th17 differentiation and IL-17 production. The authors added the explanation in the manuscript ‘discussion’ part. “A previous study showed that the increase of ROS promotes Th17 differentiation and IL-17 production through the activation of RORγt and STAT3 pathway. CoQ10 is a reactive oxygen species (ROS) scavenger. Therefore, CoQ10 can directly or indirectly regulate Th17 and IL-17 as ROS scavengers through the STAT3 pathway.”

---

## [Editor Report · Decision Letter 1]

8 Oct 2020

Liposome/gold hybrid nanoparticle encoded with CoQ10 (LGNP-CoQ10) suppressed rheumatoid arthritis via STAT3/Th17 targeting

PONE-D-20-25707R1

Dear Dr. Cho,

We’re pleased to inform you that your manuscript has been judged scientifically suitable for publication and will be formally accepted for publication once it meets all outstanding technical requirements.

Kind regards,

Yeonseok Chung

Academic Editor

PLOS ONE
---

## [Editor Report · Acceptance letter]

28 Oct 2020

PONE-D-20-25707R1 

Liposome/gold hybrid nanoparticle encoded with CoQ10 (LGNP-CoQ10) suppressed rheumatoid arthritis via STAT3/Th17 targeting 

Dear Dr. Cho:

I'm pleased to inform you that your manuscript has been deemed suitable for publication in PLOS ONE. Congratulations! Your manuscript is now with our production department. 

Kind regards, 

on behalf of

Dr Yeonseok Chung 

Academic Editor

PLOS ONE